# Self-care practice and associated factors among hypertensive patients who have follow-ups in public hospitals of Bahir-Dar City, Northwest Ethiopia, a mixed study

Gebremedhin Hailu[1], Muluken Yigezu[2]*, Hordofa Gutema[3], Lydia Melkamu[3], Natnael Kebede[4], Elias Assefa[5], Addis Hordofa[1]

1 Department of Public Health, College of Health sciences, Arsi University, Arsi, Ethiopia, 2 Department of Public Health, College of Medicine and health sciences, Dire Dawa University, Dire Dawa, Ethiopia, 3 Department of Health promotion, College of Medicine Health sciences, Bahir Dar University, Bahir Dar City, Ethiopia, 4 Department of Health promotion, School of Public health College of Medicine Health sciences, Wollo University, Dessie, Ethiopia, 5 Department of Health promotion, College of Medicine Health sciences, Mizan Tipi University, Mizan Aman, Ethiopia

* muluadambyigezu@gmail.com

## Abstract

### Background

Hypertension is defined as the sustained elevation of blood pressure to levels greater than 140/90 mmHg. It is a leading cause of cardiovascular morbidity and mortality worldwide, accounting for 13% of total deaths and 7% of the global burden of disease. This condition can be significantly reduced by strictly adhering to various self-care practices for hypertension management, including compliance with prescribed antihypertensive medications, reducing salt intake, consuming a balanced diet, avoiding harmful alcohol use, engaging in regular physical exercise, and quitting tobacco smoking.

### Objectives

The aim of this study is to determine self-care practices and associated factors, as well as to explore barriers to these practices among hypertensive patients in public hospitals in Bahir Dar city, Northwest Ethiopia.

### Methods and materials

An institution-based descriptive cross-sectional study, triangulated with qualitative research, was conducted from March 12 to April 12, 2021. A total of 415 participants were selected from three study settings. Data collected from a structured questionnaire were entered into EpiData v3.01 and analyzed using SPSS version 21. Both univariable and multivariable binary logistic regression analyses were performed. The

**Data availability statement:** All relevant data are within the paper and its Supporting Information files.

**Funding:** The author(s) received no specific funding for this work.

**Competing interests:** The authors state that they have no competing interests.

**Abbreviations:** AOR, Adjusted Odds Ratio; BP, Blood Pressure; CI, Confidence Interval; CVD, Cardio Vascular Disease; DALYs, Disability-AdjustedLlife Years; EFY, Ethiopian Fiscal Year; FHCSRH, Felege-Hiwet Comprehensive Specialized Referral Hospital; HBP-SCP, High Blood Pressure Self-Care Profile; HSC, Hypertension Self-Care; HTN, Hypertension; IDI, In Depth Interview; JNC, Joint National Committee; NCDs, Non- Communicable Diseases; OR, Odds Ratio; OSSS, Oslo Social Support Scale,TGSRH, Tibebe Gihon Specialized Referral Hospital.

statistical significance of associations between variables was determined using odds ratios with 95% confidence intervals (CIs) and p-values below 0.05. Eight purposively selected individuals participated in the qualitative component of the study.

## Results

Good self-care practices for hypertension were observed in 44.8% of participants. Significant predictors of good hypertension self-care practices included age ≥ 60, education level of secondary school or higher, employment in government positions, controlled blood pressure, good knowledge about hypertension, strong social support, and a positive perceived health status, with Adjusted Odds Ratios (AOR) and 95% Confidence Intervals (CI) of 3.04 (1.26, 7.33), 7.82 (2.79, 21.98), 1.53 (1.53, 14.90), 3.14 (1.70, 5.80), 2.27 (1.17, 4.41), 2.71 (1.31, 5.61), and 2.56 (1.35, 4.85), respectively. Major identified barriers included lack of emotional stability and stress, financial problems, lack of commitment, lack of attention, and inadequate counseling from health professionals.

## Conclusions

Less than half of the participants demonstrated good self-care practices for managing hypertension. Strategies, programs, and guidelines should be developed to help clients understand the importance of multi-dimensional well-being in relation to various aspects of hypertension self-care practices. Furthermore, all patients should receive comprehensive and tailored information on recommended self-care practices and be assessed for barriers to adherence to these practices.

### Introduction

Hypertension also called high blood pressure level is that the elevation of pressure within the arteries [1]. It is sustained elevation of the blood pressure to ≥ 140/90 mmHg [2]. It is the major risk factor for ischemic and hemorrhagic strokes, MI, heart failure, chronic kidney disease, peripheral vascular disease, cognitive decline, and premature death. It is the leading worldwide risk factor for cardiovascular disease (CVD) and mortality and is responsible for 13% of the total deaths and 7% of the global burden of disease [3].

The anticipated number of people with hypertension worldwide in 2010 was over 1.4 billion, and by 2025, that number is likely to significantly exceed 1.6 billion [1]. Hypertension is continously becoming a worldwide urgent issue. It is also a major contributor to the global burden of non- communicable diseases (NCDs). In 2016, It contributed to about 17.9 million (44%) deaths out of 40.9 million (71%) global deaths as a result of non-communicable diseases related mortalities [4]. About 40% of the world adult population lives with hypertension [5], and 28.5% are in high-income countries while 31.5% are in low-and middle-income countries [1]. Accounting for up to 54% of stroke and 47% of ischemic heart disease as well as 13.5% disability-adjusted life years (DALYs) [6].

Africa continent seems to be the most affected region in the world. About 46% for both sexes combined. The age-standardized prevalence of hypertension is 25.9% among different population groups in sub-Saharan Africa [7]. Hypertension is continued to be a public health problem in developing countries, including Ethiopia. The national overall prevalence of hypertension (HTN) in Ethiopia was 19.6% [8]. In Bahir Dar, the prevalence of hypertension among the adult population was revealed to be 25.1% [9].

Self-care is the maintenance of healthy well-being in a patient's interest by making certain day to day decisions and actions to have control over their illness [10].

The movement towards self-management of a disease has been observed in literature since the 1990s. The proceedings of the National Conference on Self-Management of Chronic Disease, which was held in 2003 in Australia, confirmed that the medical approach to treating these diseases is prescriptive and authoritarian and does not consider the subjectivity of the carrier and the importance of self-management of care [11].

Hypertension has been identified as the leading modifiable risk factor for cardiovascular disease and consequently represents a major cause of premature morbidity and mortality due to adverse cardiovascular and cerebrovascular events [12]. Blood pressure (BP) control and management of hypertension can be achieved through antihypertensive drug treatment, which has proved to be clinically effective [13]. However, recent evidence (including control rates across 12 countries) suggests that BP control through antihypertensive treatment is suboptimal, with at least 20% of that prescribed treatment failing to achieve control [14].

Hypertension is estimated to affect more than one in three adults aged 25 and over (or about one billion people) worldwide. Africa sees the highest prevalence of hypertension (46 percent of adults aged 25 and over), while the Americas the lowest (35 percent). Owing to appropriate public policies and better access to health care, high-income countries have a lower prevalence of hypertension (35 percent) than low- and medium-income countries (40 percent) [15].

Self-care is considered as a basic care for patients with chronic conditions to have a better quality of life by refraining from possibilities of disabilities and to reduce the rising health care expenditure. It has shown that it reduces primary care visits, outpatient visits by 17% and emergency department visit up to 50% [16].

However hypertensive patients often do not implement the recommended self-care practices [17] and ultimately suffer from uncontrolled blood pressure. According to a recent study, about 52.5% and 50% of hypertensive patients in Ayder hospital, Mekelle northern Ethiopia [18], and southwest Ethiopia live with uncontrolled blood pressure respectively [19].

Therefore, the aim of this study was to assess the extent and determinants of hypertension self-care practices, taking into account identified deficiencies, and to investigate the obstacles faced by patients receiving follow-up care at public hospitals in Bahir Dar, Ethiopia. The results of this study will assist healthcare professionals in creating guidelines for self-care management, educating hypertensive patients, enhancing their compliance, and improving communication to positively impact self-care behaviors. Furthermore, these findings will be instrumental in shaping health policies and establishing self-care management programs for hypertensive patients across various healthcare facilities to prevent hypertension-related complications.

## Materials and methods

### Study design, setting and period

An institution-based cross-sectional study triangulated with a qualitative study was applied in public hospitals of Bahir Dar city, Northwest Ethiopia from March 12,2021 to April 12, 2021. The city has three public hospitals in which the two are specialized and the other one is a primary hospital. These hospitals are namely Felege-hiwet comprehensive specialized referral hospital (FHCSRH), Tibebe Gihon specialized referral hospital (TGSRH), and Adis Alem primary hospitals. Bahir-Dar is located approximately 578 km north-northwest of Addis Ababa. According to Ethiopian fiscal year (EFY) of 2012, the city has around 324,323 populations, of which 51.3% (166,388) of them are females and the remaining 48.7% (157,945) are males [20]. On average; Felege hiwot, Tibebe Gihon, and Adis-Alem hospitals serve approximately 408,

130, and 235 hypertensive clients each month respectively[20]. On average, Felege hiwot, Tibebe Gihon, and Adis-Alem hospitals each serve approximately 408, 130, and 235 hypertensive clients per month, respectively.

## Study participants

The source populations for this study were all hypertensive patients who had follow-ups in public hospitals in Bahir Dar city, with the study population including all selected hypertensive patients who had follow-ups in public hospitals in Bahir Dar city at the time of data collection.

## Eligibility criteria

All hypertensive patients aged 18 and older who had follow-ups at public hospitals in Bahir Dar city were included in this study. However, patients who were critically ill or had cognitive impairments during the data collection period were excluded. Additionally, patients with follow-up periods of less than six months were excluded, as a certain activity or behavior is assumed to be established only if it is practiced repeatedly for at least six months.

## Sample size determination and sampling procedures

The sample size for the quantitative component was calculated using the formula for a single population Proportion with a 95% confidence level, 5% margin of error, and a prevalence of poor hypertension self-care practice of 51% [21]. This yielded a required sample size of 385, rounded up to 423 after accounting for a 10% non-response rate.

For the qualitative component, 8 individuals were involved in the in-depth interviews based on the principle of saturation in qualitative research [22].

**Sampling procedure.** For the quantitative component, a total of 423 study participants were chosen through systematic random sampling. These participants were distributed proportionally across the study areas, with the required number selected systematically, involving every other patient. Initially, card numbers of patients scheduled to visit health facilities during the data collection period were used as a sampling frame. Subsequently, the first participant was randomly selected from this list of cards, and then every other patient was interviewed to gather the necessary data.

In the qualitative part of the study, participants were selected using quota-based purposive sampling through a heterogeneous sampling approach. This method involved selecting clients for in-depth interviews (IDIs) based on factors such as age, sex, place of residence, duration of illness, and the presence or absence of other comorbidities.

## Study variables

The dependent variable was Hypertension self-care practice and independent variables were Socio-demographic variables (Age, sex, marital status, place of residence, educational status, and occupation), Empowerment factor (self-efficacy), Knowledge, Social support, Perceived health status, and other health-related factors (illness duration, family history of HBP, source of information related to HBP, availability of a place to make exercise, and presence of co-commodities).

## Operational definitions

**Hypertension self-care practice:** was measured by the 20-item measure of Hypertension Self-Care Profile (HBP-SCP) and those who scored above the mean are considered to have favorable/good self-care practices [21].

**Knowledge:** was measured by the overall summation of 12 items of true/false bases after reverse coding of the negatively worded items and those respondents who scored above the mean were considered as having good knowledge towards hypertension and its self-care practices [23].

**Perceived social support:** was measured by the 3-item measure of Oslo social support scale (OSSS-3) and participants are classified as having low, moderate, and strong social support [24].

**Self-efficacy:** was measured by summing of results 5-item measures which were rated from 1–10 and those respondents who scored mean and above were considered to have high self-efficacy [25].

**Perceived health status:** was measured by summing of results 12-item measures of the short form (SF-12) health survey and participants who scored mean and above were considered to have good perceived health status [26].

## Data collection tools and procedures

For the quantitative aspect, structured face-to-face interviews were conducted using a questionnaire developed from relevant literature to evaluate the self-care practices of hypertensive patients and the factors associated with these practices. The questionnaire, initially in English, was translated into Amharic. It comprised four main sections:

Socio-demographic variables: This section assessed age, gender, marital status, educational background, and occupation. Health profile characteristics: This section covered aspects such as family history of high blood pressure, measured blood pressure levels, sources of hypertension-related information, duration of illness, presence of other medical conditions, and access to facilities for physical exercise. Hypertensive self-care practices: The participants' self-care practices were evaluated using the Behavioral Scale of the Hypertension Self-Care Profile (HBP-SCP), a 20-item scale where responses ranged from 'not at all' (1) to 'always' (4) [27]. Factors associated with hypertensive self-care practices were assessed in the fourth section.

The principal investigator recruited four data collectors who were supervised by the principal investigator and two additional supervisors. An in-depth interview (IDI) was conducted to investigate the barriers preventing clients from engaging in effective self-care practices to manage their condition. The semi-structured, open-ended questionnaire used in the IDI was developed by the principal investigator and included questions based on 'how' and 'why'. During data collection, the principal investigator utilized a digital audio recorder to document the information gathered.

## Data quality control

Prior to data collection, the questionnaire underwent a pretest on 5% of the sample at Han Health Center, leading to necessary modifications. The pretest data was not included in the final analysis. Data collectors and supervisors received training on questionnaire content and privacy protection. Daily reviews ensured completeness, accuracy, and clarity of the questionnaires.

As to the qualitative part, the rigor and trustworthiness of the study was ensured by considering the criteria of credibility, dependability, conformability, and transferability [28]. To ensure credibility, data were collected from respondents with different backgrounds. To ensure dependability, accurate documentation was maintained by minimizing spelling errors through frequent checks. The analyzed and interpreted data underwent continuous peer review. Conformability was achieved by incorporating quotes from participants to link their words with the findings. Both quantitative and qualitative data are stored securely to maintain confidentiality and for data backup when needed.

## Data processing and analysis

For the quantitative part, descriptive statistics such as frequencies, percentages, mean values, and standard deviations were computed for respondent characteristics and other measured study variables. The outcome variable, H-SCP, was dichotomized into good and poor based on the analyzed mean scores. Binary logistic regression was conducted to examine the associations between the outcome variable and each of the explanatory variables. The statistical significance of associations between variables was determined using odds ratios with a 95% confidence interval (CI) and p-values below 0.05. Subsequently, independent variables with less than or equal to 0.2 were selected to be candidates for multivariable logistic regression [21]. For the qualitative part, the researcher first managed the data by creating and organizing files through data collection, transcription, and translation. Subsequently, the translated data were read and reread until the full meaning of the contents was understood. Codes were attached to each quote from the participants, and the data were

displayed to capture the variation or richness of each code. Data reduction was then performed to distill the information to extract the most essential concepts and relationships, and finally, the data was interpreted. To facilitate this analysis, Atlas. ti 7 software was used.

Quantitative and qualitative data were collected simultaneously, analyzed side by side, and ultimately, the results from each data source were combined to provide meaning [22].

## Ethical consideration

In order to conduct this research, the authors tried to address the Declaration of Helsinki Ethical principles for medical research. Ethical approval was obtained from the Ethical committee of Bahir Dar University, College of Medicine and Health Sciences from the office of chief academic and research directorate on February 26, 2021 and the approval letter was registered under protocol number of 0126/2021. Informed voluntary written and signed consent was obtained from all study participants prior to start data collection. For both the quantitative and qualitative parts, the interviewees were informed about the objectives and data collection procedures. For qualitative part, the IDI was conducted in a separate area from the follow-up room after the selected patient completed his/her treatments. The participants were allowed to consider their participation and they were allowed to withdraw from the study when they wished to do so. Participant's name or personal identifier wasn't included in the IDI and structured interviews to ensure participants' confidentiality. All transcripts and other data were/are kept in a locked file.

## Results

### Socio-demographic characteristics of the study participants

From a total of 423 sampled participants, 415 hypertensive patients who had follow-ups in public hospitals of Bahir Dar city participated in the study, resulting in a response rate of 98.1%. Out of the total respondents, 213 (51.3%) were females (Table 1). The participants in the study ranged in age from a minimum of 25 years to a maximum of 89 years, with a mean age of 53.52 ± 13.7 years (standard deviation).

The majority of the participants were married, accounting for 234 (56.4%). Of all the respondents, 276 (66.5%) were residents of urban areas. Regarding educational status, 149 (35.9%) and 113 (27.2%) of the participants were unable to read and write and attended college and above, respectively. One hundred eight (26%) of the respondents were government employees. (Table 1).

### Health profile related characteristics of the study participants

Out of 415 respondents, 157 (37.8%) had a family history of hypertension. All participants had access to health education. The average duration of illness of the respondents was 5.98 ± 4.404 years. One hundred forty-two (34.2%) of the respondents had at least one possible comorbidity. Only 81 (19.5%) of the study participants had an available place for physical exercise. The mean systolic and diastolic blood pressure of the participants were 138.2 and 85.3 mmHg, respectively. (Table 2).

### Hypertension self-care practice of the study participants

The overall mean of hypertension self-care practice was 52.9 ± 10.7. Among the 415 study participants, 186 (44.8%) had good hypertension self-care practice with a confidence interval (CI) of (40.0, 49.6) at a 5% level of significance. Among all the respondents involved in the study, 112 (27%) of them never engaged in regular physical exercise. 129 (31.1%) and 158 (38.1%) of the participants were practicing the consumption of less than 1 teaspoon of table salt per day always and most of the time, respectively. The majority of the participants never checked their blood pressure at home, accounting for 289 (69.6%). The majority of the respondents practiced non-smoking, accounting for 378 (91.1%). (Table 3).

**Table 1. Socio-demographic characteristics of respondents for quantitative study in public hospitals of Bahir Dar city, North West Ethiopia June, 2021 (n = 415).**

| Variables | | Number | Percent |
|---|---|---|---|
| Sex | Male | 202 | 48.7 |
| | Female | 213 | 51.3 |
| Age<br>Mean = 53.52 ± 13.7 SD | <=40 | 82 | 19.8 |
| | 41-59 | 183 | 44.1 |
| | >=60 | 150 | 36.1 |
| Marital status | Single | 45 | 10.8 |
| | Married | 234 | 56.4 |
| | Divorced | 67 | 16.1 |
| | Widowed | 69 | 16.6 |
| Residence | Urban | 276 | 66.5 |
| | Rural | 139 | 33.5 |
| Education | 'Unable To Read And Write' | 149 | 35.9 |
| | 'Able To Read And Write' | 53 | 12.8 |
| | 'Primary School' | 46 | 11.1 |
| | 'Secondary School' | 54 | 13.0 |
| | 'College or Above' | 113 | 27.2 |
| Occupation | Farmer | 82 | 19.8 |
| | Government Employee | 108 | 26.0 |
| | Private Employee | 84 | 20.2 |
| | Daily Laborer | 12 | 2.9 |
| | House wife | 76 | 18.3 |
| | Other | 53 | 12.8 |

## Factors associated with Hypertension self-care practice of the study participants

In the multivariable logistic regression process, only age, educational status, occupational status, hypertension status, knowledge, social support, and perceived health status were significant predictors. The model containing the best predictors was fitted to explain the factors determining good self-care practices (Hosmer-Lemeshow statistic = 0.372). (Table 4).

In this study, participants aged 41–59 and sixty and above were about 3 and 2.3 times more likely to have good H-SCP than the younger population aged less than or equal to forty (AOR = 2.32, 95% CI = 1.03, 5.20) and (AOR = 3.22, 95% CI = 1.29, 8.06) respectively. Respondents who attended primary school, secondary school, and college and above were about 2.9, 5.9, and 7.8 times more likely to have good hypertension self-care practices compared to those respondents who can't read and write, with (AOR = 2.91, 95% CI = 1.35, 6.29), (AOR = 5.90, CI = 2.10, 16.58), and (AOR = 7.82, 95% CI = 2.79, 21.98), respectively. Regarding occupational status, government employees were about 4.8 times more likely to engage in good self-care practices for hypertension (AOR = 4.77, 95% CI = 1.53, 14.90). (Table 4).

During the study period, blood pressure status was identified as a significant predictor of self-care practices for hypertension. Participants with controlled blood pressure, defined as a systolic blood pressure of less than 140 mmHg and a diastolic blood pressure of less than 90 mmHg, were approximately 3.1 times more likely to exhibit favorable hypertension self-care practices compared to those with uncontrolled blood pressure (≥ 140/90 mmHg) (AOR = 3.14, 95% CI = 1.70, 5.80). (Table 4).

Participants who possessed good knowledge about hypertension and its self-management practices were about 2.3 times more likely to engage in favorable hypertension self-care practices (AOR = 2.27, 95% CI = 1.17, 4.41). (Table 4).

**Table 2.** *Health profile related characteristics of respondents in public hospitals of Bahir Dar city, North West Ethiopia June, 2021 (n=415).*

| Variables | | Number | Percentages |
|---|---|---|---|
| Family history of HBP | Yes | 157 | 37.8 |
| | No | 258 | 62.2 |
| Source of health information | Books | 37 | 8.9 |
| | Magazines | 8 | 1.9 |
| | Electronic media | 123 | 29.6 |
| | Health education | 415 | 100.0 |
| Duration of illness; mean=5.98±4.404, Minimum=1 yr, maximum=22 yrs. | | | |
| Place for physical exercise | Yes | 88 | 21.2 |
| | No | 327 | 78.8 |
| Presence of comorbidities | Diabetes mellitus | 88 | 21.2 |
| | Chronic kidney disease | 25 | 6.0 |
| | Chronic heart failure | 31 | 7.5 |
| | Stroke | 11 | 2.7 |
| | Overall | 142 | 34.2 |
| Measured SBP(in mmhg) | Mean=138.23±12.105 | | |
| Measured DBP(in mmhg) | Mean=85.29±7.576 | | |

Social support was also identified as a significant predictor of self-care practices among hypertensive clients. Participants with strong social support were approximately 2.7 times more likely to exhibit good hypertension self-care practices compared to those with low social support (AOR=2.71, 95% CI=1.31, 5.61). (Table 4).

Participants who reported good perceived health status were about 2.6 times more likely to engage in good hypertension self-care practices compared to those with poor perceived health status (AOR=2.56, 95% CI=1.35, 4.85). (Table 4).

## Qualitative results

### Socio-demographic characteristics of the study participants

In the qualitative part, eight purposively selected hypertensive clients who were not included in the quantitative study participated in the in-depth interviews (IDIs). The interviews lasted an average of 15 minutes. Five of the respondents were male, and six were married, with two having comorbidities. (Table 5).

### Barriers of hypertension self-care practices of the study participants

After collecting data in Amharic through in-depth interviews, word-for-word transcription was conducted, translated into English, read and reviewed. Atlas ti.7 software was used to analyze the data, and three themes were explored among the in-depth interviews of the eight study participants. These themes are listed and discussed as follows: individual barriers, health facility barriers, social and/or economic barriers

**Individual level barriers of the study participants.** One of the main barriers of hypertension self-care practices at individual level was lack of knowledge/awareness about how to manage their condition. Practices which were repeatedly explained by the respondents were only as they should take medications regularly; restrict salt during preparing their foods; and limit excessive consumption of alcohol.

*"the only thing I know to control my condition [hypertension] is to take medications ordered by the doctors and I do know nothing else" (29 years old, female, P4)*

**Table 3.** *Distributions of hypertension self-care practices of hypertensive patients in public hospitals of Bahir Dar city, North West Ethiopia June, 2021 (n = 415).*

| Practices | Frequencies (%) | | | |
|---|---|---|---|---|
| | **Always** | **Often** | **Sometimes** | **Not at all** |
| Take part regular physical activity (e.g., 30 minutes of walking 4–5 times per week)? | 73(17.6) | 107(25.8) | 123(29.6) | 112(27) |
| Read nutrition facts label to check information on sodium content? | 55(13.3) | 94(22.7) | 66(15.9) | 200(48.2) |
| Eat low-salt foods (e.g., fresh vegetables)? | 131(31.6) | 192(46.3) | 85(20.5) | 7(1.7) |
| Limit the use of high-salt condiments? | 99(23.9) | 131(31.6) | 127(30.6) | 58(14) |
| Eat less than 1 teaspoon of table salt per day? | 129(31.1) | 158(38.1) | 77(18.6) | 51(12.3) |
| Avoid consuming fatty foods? | 70(16.9) | 193(46.5) | 139(33.5) | 13(3.1) |
| Eat fewer foods that are high in fat (e.g., red meat, butter)? | 78(18.8) | 162(39) | 138(33.3) | 37(8.9) |
| Replace traditional high-fat foods (e.g., deep-fried chicken) with low-fat products (e.g., baked chicken)? | 40(9.6) | 70(16.9) | 126(30.4) | 179(43.1) |
| Use bake or steam instead of frying when cooking? | 95(22.9) | 108(26) | 81(19.5) | 131(31.6) |
| Read the nutrition label to check info on fat products (e.g., butter, red meat)? | 60(14.5) | 77(18.6) | 71(17.1) | 207(49.9) |
| Eat 5 or more servings of fruits and vegetables daily? | 22(5.3) | 20(4.8) | 153(36.9) | 220(53) |
| Practice moderation in drinking alcohol daily (2 glasses or less for men; 1 glass or less for women)? | 187(45.1) | 69(16.6) | 69(16.6) | 90(21.7) |
| Practice non-smoking? | 378(91.1) | 8(1.9) | 13(3.1) | 16(3.9) |
| Check your blood pressure at home? | 89(21.4) | 22(5.3) | 15(3.6) | 289(69.6) |
| Remember to take your blood pressure medicine? | 251(60.5) | 136(32.8) | 6(1.4) | 22(5.3) |
| Remember to fill your prescriptions? | 331(79.8) | 56(13.5) | 15(3.6) | 13(3.1) |
| Keep your weight down? | 77(18.6) | 121(29.2) | 100(24.1) | 117(28.2) |
| Monitor situations that cause a high level of stress (e.g., arguments, death in the family) resulting in blood pressure elevation? | 71(17.1) | 161(38.8) | 86(20.7) | 97(23.4) |
| Engage in activities that can lower stress (e.g., deep breathing, meditation? | 140(33.7) | 196(47.2) | 52(12.5) | 27(6.5) |
| See a doctor regularly? | 140(33.7) | 196(47.2) | 52(12.5) | 27(6.5) |
| **Overall mean** | **52.87** | | | |
| Scored | Above mean | 186(44.8%) | | |
| | Below mean | 229(55.2%) | | |

Lack of emotional stability and stress was also stated as a main barrier to the performance of self-management of hypertension. *".... Being emotional is often the thing that really hurts me because things like this happen out of my control. I understand that I have this problem as a problem, but even I try not to do that it really happens and this makes me not to care to my condition and consequently sometimes leads my blood pressure even to worsen". (*50 years old, male, P1)

Another client who was a prisoner in Sebatamit maremiya bet and following his treatment in Tibebe Gihon hospital stated about the stress which was leading him to feel not confortable or not to follow the recommended H-SCP. He said that

*"Sometimes stress at me, especially if I don't find my family,…because my families are living at rent and when I watch my family suffering as a result of increased costs of expenditure… I got stressed and even sometimes forget taking my medications and doing physical exercises which I usually perform with my other prison mates". (*55 years old, male, P7).

**Table 4.** *Bivariable and multivariable logistic regression showing actors associated with self-care practices of hypertensive patients in Public hospitals of Bahir Dar city, North West Ethiopia June, 2021 (n = 415).*

| Variable | Categories | Self-care practice | | COR at 95% CI | AOR at 95% CI | P-value |
|---|---|---|---|---|---|---|
| | | Good(n) | Poor(n) | | | |
| Gender | Male | 106 | 96 | 1 | 1 | |
| | Female | 80 | 133 | 0.55 (0.37, 0.81) | 0.84 (0.45, 1.57) | 0.576 |
| Age | ≤40 | 46 | 36 | 1 | 1 | |
| | 41-59 | 107 | 102 | 0.77 (0.46, 1.31) | 2.32 (1.03, 5.20) | 0.042 |
| | ≥60 | 33 | 91 | 0.38 (0.22, 0.66) | 3.04 (1.26, 7.33) | 0.013 |
| Marital status | Single | 27 | 18 | 1 | 1 | |
| | Married | 132 | 102 | 0.86 (0.45, 1.65) | 1.77 (0.62, 5.09) | 0.287 |
| | Divorced | 16 | 51 | 0.21 (0.09, 0.48) | 1.01 (0.30, 4.44) | 0.986 |
| | Widowed | 11 | 58 | 0.13 (0.05, 0.30) | 0.86 (0.23, 3.23) | 0.825 |
| Residence | Urban | 147 | 129 | 1 | 1 | |
| | Rural | 39 | 100 | 0.34 (0.22, 0.53) | 1.33 (0.54, 3.25) | 0.537 |
| Educational status | Illiterate | 23 | 126 | 1 | 1 | |
| | Primary school | 34 | 65 | 2.86 (1.56, 5.26) | 2.91 (1.35, 6.29) | 0.006 |
| | Secondary school | 32 | 22 | 7.97 (3.95, 16.1) | 5.90 (2.10, 16.58) | 0.001 |
| | College/above | 97 | 16 | 33.2 (16.6, 66.3) | 7.82 (2.79, 21.98) | 0.000 |
| Occupation | Farmer | 18 | 64 | 1 | 1 | |
| | Government Employee | 96 | 12 | 28.4 (12.83, 63) | 4.77 (1.53, 14.90) | 0.007 |
| | Private Employee | 33 | 51 | 2.30 (1.16, 4.55) | 1.36 (0.53, 3.52) | 0.522 |
| | Housewife | 16 | 60 | 0.95 (0.44, 2.03) | 0.61 (0.24, 1.60) | 0.316 |
| | Other | 23 | 42 | 1.36 (0.72, 2.58) | 1.94 (0.73, 5.16) | 0.185 |
| Family Hx of HBP | Yes | 93 | 64 | 1 | 1 | |
| | No | 93 | 165 | 0.39 (0.26, 0.58) | 1.02 (0.52, 1.94) | 0.995 |
| Sources of information | Health education | 100 | 166 | 1 | 1 | |
| | HE & others | 86 | 63 | 2.27 (1.51, 3.41) | 0.72 (0.37, 1.39) | 0.323 |
| Duration of illness | < 5 years | 92 | 92 | 1 | 1 | |
| | ≥ 5 years | 94 | 137 | 0.69 (0.46, 1.01) | 1.24 (0.61, 2.52) | 0.549 |
| Place of exercise | Yes | 27 | 202 | 1 | 1 | |
| | No | 54 | 132 | 0.43 (0.26, 0.69) | 0.97 (0.46, 2.06) | 0.941 |
| BP control | Uncontrolled | 103 | 174 | 1 | 1 | |
| | Controlled | 83 | 55 | 2.55 (1.68, 3.88) | 3.14 (1.70, 5.80) | 0.000 |
| Knowledge | Poor | 24 | 130 | 1 | 1 | |
| | Good | 162 | 99 | 8.86 (5.37, 14.6) | 2.27 (1.17, 4.41) | 0.015 |
| Self-efficacy | Low | 34 | 143 | 1 | 1 | |
| | High | 152 | 86 | 7.43 (4.7, 11.75) | 0.92 (0.42, 1.99) | 0.825 |
| Perceived health status | Poor | 31 | 132 | 1 | 1 | |
| | Good | 155 | 97 | 6.8 (4.27, 10.85) | 2.56 (1.35, 4.85) | 0.004 |
| Social support | Low | 37 | 118 | 1 | 1 | |
| | Moderate | 68 | 70 | 3.1 (1.88, 5.1) | 1.65 (0.82, 3.33) | 0.055 |
| | Strong | 81 | 41 | 6.3 (3.72, 10.67) | 2.71 (1.31, 5.61) | 0.007 |

**Table 5. Socio-demographic characteristics of clients involved in qualitative study of H-SCP, in public hospitals of Bahir Dar city, North Western June, 2021 (n = 415).**

| S. no | Code | Sex | Age | Institution | Residence | Duration of illness (month) | Marital status | Occupational status | Educational status | Comorbidities |
|---|---|---|---|---|---|---|---|---|---|---|
| 1. | P 1 | M | 50 | FHCRH | Urban | 24 | Married | Gov.t employee | MA | Dm |
| 2. | P 2 | M | 65 | FHCRH | Urban | 15 | Married | Government employee/ lowyer | 12 + 1 | No |
| 3. | P 3 | F | 48 | FHCRH | Urban | 9 | Married | Govornment employee/ police | 12 | No |
| 4. | P 4 | F | 29 | FHCRH | Rural | 3 | Married | Farmer | Illiterate | No |
| 5. | P 5 | M | 80 | AA1°H | Rural | 2 | Divorced | Farmer | Illiterate | CKD |
| 6. | P 6 | M | 60 | TGSRH | Urban | 1 | Married | Government employee/ Teacher | 12 + 4 | CHF |
| 7. | P 7 | M | 55 | TGSRH | Urban | 2 | Married & separated | Prisoner | Diploma | No |
| 8. | P 8 | F | 25 | TGSRH | Rural | 7 | Single | Farmer | illiterate | No |

Lack of committement and determination towards different self-care practices was also another identified individual level barrier.

*"…and it takes determination. Sometimes laziness, sometimes getting up in the morning… and doing physical exercise is not comfortable". (*65 years old, male, P2).

**Institutional barriers of the study participants.** Under this theme such barriers of hypertension self-care practices discussed by/with the clients include lack of attention from health professionals and improper councelling of patients. In support of this perspective, a participant narrated that

*"we meet doctors once every three months or every four months… Doctors are also uncomfortable for us[patients]. In one room, four doctors and four clients are examined together. It is not easy for the client to express his heart, but other than that, as a personal friend, I have many medical friends and relatives, so I have not had so many problems". (A 50 years old, male, P1)*

**Social and/or economic barriers of the study participants.** Such barriers include lack of family support, peer pressure, finanitial problems, problems related to the nature of their work.

Pressures that came from friends including colleagues and neighbors were stated as contributing barriers to not perform different self-care practices.

*"… I have a problem with food and drinks. When I tell them [his friends] that I shouldn't eat and/or drink a certain foods/ drinks they try to convince me by saying "just it is for only one or two days and it will not have that much effect on your health." Again when I resist to this extent they label me as (ante degmo kifu amel alebh). Then finally in order not to offend my friends, sometimes I try to chill with them."* …" (a 50 year old, male, P1).

Financial problems like inadequate money to buy vegitables and/or fruits and sport equipments which are helpful to facilitate physical exercise and that are recommended by the professionals were also another barriers explored at this level.

*"…it is difficult to eat vegitables and fruit five time per day. Because the increased cost of expenditure. If you see the price for 1kg of banana is 40 ETB and for 1kg of orange is 80 ETB. Even if I buy such fruits one or twice a week or a month, there are children in the house and they need it. So instead of eating myself I usually give it to the children". (*60 years old, male, P6)

*"… But to do regular physical exercise, sports equipment need to be used. It takes places to do it and this might be difficult as I can't afford for it". (*65 years old, male, P2)

### Reliability of the instrument

A reliability analysis was conducted to assess the internal consistency of the measurements in this study. Individual Cronbach's Alpha values were computed for each latent variable as well as for the overall items of the tool. The overall Cronbach's Alpha, which included 70 items, was found to be 0.902. The reliability scores for the latent variables are also presented: Self-efficacy had the highest reliability score (α=0.945), followed by Perceived health status (α=0.912), while Social support had the lowest reliability score (α=0.766) (Table 6).

## Discussion

The finding of this study revealed that, the overall mean of hypertension self-care practices to be 52.87% and good hypertension self-care practices was found in 44.8% with 95 CI being between 40.0 and 49.6. This finding goes in line with studies conducted in Dessie [21] and lagos, Nigeria[29] in which good hypertension self-care practice was found to be 47.4% and 48.7% respectively. This finding is higher than the studies done in Mekelle [17], Addis Ababa [30], south Ethiopia [31], Nigeria [32], and Singur (West Bengal) [33]. On the contrary this finding is lower than the findings of such studies done in Gondar [23], and india[34]. This incongruence might be mainly due to discrepancy of health related information being provided and/or received, and the differences in the tools that were utilized to assess the outcome variable in question. The increased level of uncontrolled hypertension may also explain for the low magnitude of good self-care.

The study also found that the socio-demographic variables specifically Age, educational status, and occupation to be significant predictors of hypertension self-care practice. Participants whose age was between 41–59 and greater than or equal to 60 were around 2.3 and 3.2 times more likely to engage in favorable self-care practices as compared to younger participants whose age was less than or equal to 40. This finding goes in line with a study done in Addis Ababa public hospitals [30], Ghana [35], and Israel [36]. On the other hand this finding is in contradiction with studies done in Mekelle [17], South Ethiopia [31], and India [34] in which older clients were more likely to be found with poor self-care practices and recommended life style modifications to control hypertension. This discrepancy might be due to the differences in participants' age categorization and sample size.

**Table 6. Reliability test of tools utilized to assess Hypertension self-care practice and associated factors among hypertensive patients who have followups in public hospitals of Bahir Dar city, North West Ethiopia, June 2021.**

| S.No | Variables | Number of items | Chronbach's alpha |
|------|-----------|-----------------|-------------------|
| 1. | Hypertension self-care practice | 20 | 0.879 |
| 2. | Knowledge | 12 | 0.789 |
| 3. | Social support | 3 | 0.766 |
| 4. | Self-efficacy | 5 | 0.945 |
| 5. | Perceived health status | 12 | 0.912 |
| 6. | Overall | 52 | 0.946 |

Respondents who attended primary school, secondary school, and college and above were about 2.9, 5.9, and 7.8 times more likely to have good hypertension self-care practices as compared to those respondents who can't read and write respectively. This findings goes in line with studies done in Mekelle [17], and Dessie [37] in which participants who had an educational status of college and above had 4.21, 4.85, more good self-care practice than those who cannot read and write. The findings are also consistent with studies conducted in Ghana [35], India [33], China [38], and Saudi Arabia [39]. On the other hand, results from a study done in south Ethiopia seemed contradictory in which clients with no formal education were two (2) times more likely to practice the recommended life style modifications [31].

In terms of occupational status, government employees were approximately 3.4 times more likely to engage in favorable self-care practices for hypertension compared to farmers. This could be attributed to the higher level of knowledge among government employees regarding hypertension self-care management practices in the study.

Participants with controlled blood pressure (BP < 140/90 mmHg) were about 3.1 times more likely to have favorable hypertension self-care practices compared to those with uncontrolled blood pressure (≥140/90 mmHg). This finding is consistent with a study conducted in Mekelle, where clients with controlled blood pressure were 2.73 times more likely to have good self-care practices[17]. This result suggests that hypertension can be managed and reduced through good self-care practices.

The research identified knowledge as a crucial determinant of effective self-care practices for individuals managing hypertension. Participants who possessed a strong understanding of hypertension and its self-management strategies were found to be approximately 2.3 times more likely to adopt positive self-care behaviors. This highlights the importance of education in empowering individuals to take control of their health. Further qualitative analysis indicated that a significant barrier to effective self-care was the lack of knowledge regarding hypertension management. For instance, many participants expressed uncertainty about the recommended daily salt intake, as well as the types and amounts of physical exercise necessary for maintaining healthy blood pressure levels. This gap in knowledge underscores the need for targeted educational interventions to enhance self-management skills among hypertensive patients.

The quantitative findings regarding the relationship between knowledge and self-care practices in this study indicate a weaker association compared to results from previous research conducted in various regions of Ethiopia. Specifically, studies in Mekelle [17], South Ethiopia [31], and Addis Ababa [40] found that individuals with good knowledge of hypertension management were associated with a likelihood of engaging in good self-care practices that was approximately 6.19 times, 6.19 times, and 13 times higher, respectively. These findings suggest a strong correlation between knowledge and self-care behaviors in those populations. In contrast, the current study's results indicate a lower association, suggesting that while knowledge is important, its impact on self-care practices may vary by region and population.

On the other hand, the findings from this study show a stronger association than those observed in a study conducted at Dessie Teaching and Referral Hospital [37], where good knowledge was associated with a likelihood of approximately 1.8 times for engaging in effective self-care practices. This comparison highlights the variability in the influence of knowledge on self-care across different settings and populations.

In addition to knowledge, the study also revealed that social support plays a vital role in the self-care practices of individuals with hypertension. Participants who reported having substantial social support were approximately 2.7 times more likely to engage in effective self-care routines compared to those who experienced inadequate support. This finding was corroborated by qualitative data, where participants shared experiences of challenges posed by friends, colleagues, and family members. For example, some clients noted difficulties in adhering to dietary recommendations, such as consistently preparing meals with the appropriate amount of salt. They also faced social pressures that encouraged unhealthy behaviors, including alcohol consumption and the intake of high-fat foods.

Interestingly, these findings contrast with a previous study conducted in Addis Ababa, which suggested that societal support was strongly linked to adherence to lifestyle modifications among hypertensive individuals. In that particular study, respondents who received social support were found to be nearly 11 times more likely to adhere to recommended lifestyle

changes[30]. The discrepancy between the two studies may stem from differences in the methodologies employed by researchers to evaluate social support. Variations in the tools and frameworks used to measure social support could account for the differing outcomes observed, highlighting the complexity of this factor in the context of hypertension management.

Additionally, the study revealed that clients who perceived their health status as good were about 2.6 times more likely to engage in favorable hypertension self-care practices compared to those who perceived their health status as poor. This finding aligns with the notion that positive health perceptions can motivate individuals to adopt healthier behaviors. However, this result contrasts with a study conducted in West Bengal, India [33], which reported no significant association between perceived health status and hypertension self-care practices. The discrepancy between these findings may be attributed to several factors, including differences in the socio-demographic characteristics of the respondents, variations in the tools used to assess perceived health status, and the relatively small sample size of only 124 participants in the Indian study.

Interestingly, this quantitative finding contradicts qualitative insights gathered from participants in the current study. Many individuals expressed that experiencing poor day-to-day health prompted them to reevaluate and modify their self-care practices. This suggests that a negative perception of health can serve as a catalyst for self-reflection and may encourage individuals to identify and address gaps in their management strategies for hypertension. This divergence between quantitative and qualitative findings emphasizes the complexity of health perceptions and their impact on self-care behaviors, indicating that both positive and negative health perceptions can influence individuals' approaches to managing their condition.

## Limitations of the study

The potential for recall bias in relation to the time of initiation, duration, and patterns of hypertension self-care practices is a significant consideration in this study. Participants may have difficulty accurately recalling when they started engaging in these practices, how long they have been following them, and the specific patterns they have adopted. This could lead to inaccuracies or inconsistencies in the data collected, affecting the overall reliability and validity of the findings.

Moreover, the presence of social desirability bias is another important factor to take into account. Since the self-care practices reported by the study participants are based on their own self-reports, there is a possibility that they may have provided responses that they believe are socially acceptable or desirable, rather than reflecting their actual behaviors. This could potentially lead to an overestimation of the adherence to self-care practices or a distortion of the true picture of their engagement in these activities.

## Conclusions

The findings of this study reveal that a relatively low percentage of participants, only 44.8%, engaged in good hypertension self-care practices. This suggests that there is a significant portion of the population who may not be effectively managing their hypertension. The predictors of hypertension self-care practices identified in the study, including age, educational status, occupational status, hypertension status, knowledge, social support, and perceived health status, highlight the complex interplay of various factors that influence individuals' engagement in self-care behaviors.

The barriers and challenges identified by participants shed light on the multifaceted nature of the obstacles they face in adhering to hypertension self-care practices. Factors such as lack of knowledge about how to manage their condition, emotional instability and stress, lack of commitment, lack of family support, peer pressure, financial problems, barriers related to their nature of work, and inadequate attention and counseling from health professionals all contribute to the difficulties individuals encounter in maintaining optimal self-care.

These findings underscore the importance of addressing not only individual-level factors but also broader social and environmental influences on hypertension self-care practices. Interventions aimed at improving self-care behaviors should

consider these barriers and tailor strategies to provide education, support, and resources to help individuals overcome these challenges. By addressing the underlying factors that hinder effective self-care, healthcare providers and policymakers can better support individuals in managing their hypertension and improving their overall health outcomes.

## Supporting information

**S1 File. SPSS Data.**
(SAV)

## Acknowledgments

The Authors would like to thank Bahir Dar University College of Medicine and Health Science Department of Health Promotion and Behavioral science for coordination of the overall research activity. The authors would also acknowledge the data collectors, supervisors, and all personnel who were involved in the accomplishment of this study.

## Author contributions

**Conceptualization:** Muluken Yigezu, Iydia Melkamu, Natnael Kebede.

**Data curation:** Hordofa Gutema, Natnael Kebede, Elias Assefa, Gebremedhin Hailu.

**Formal analysis:** Muluken Yigezu, Iydia Melkamu, Gebremedhin Hailu.

**Funding acquisition:** Hordofa Gutema, Natnael Kebede, Elias Assefa, Addis Hordofa.

**Investigation:** Muluken Yigezu, Iydia Melkamu, Natnael Kebede, Addis Hordofa, Gebremedhin Hailu.

**Methodology:** Muluken Yigezu, Hordofa Gutema, Elias Assefa, Gebremedhin Hailu.

**Project administration:** Muluken Yigezu, Iydia Melkamu, Natnael Kebede, Elias Assefa, Gebremedhin Hailu.

**Resources:** Hordofa Gutema, Natnael Kebede, Elias Assefa, Addis Hordofa, Gebremedhin Hailu.

**Software:** Muluken Yigezu, Gebremedhin Hailu.

**Supervision:** Muluken Yigezu, Hordofa Gutema, Iydia Melkamu, Natnael Kebede, Elias Assefa, Addis Hordofa, Gebremedhin Hailu.

**Validation:** Muluken Yigezu, Elias Assefa, Addis Hordofa, Gebremedhin Hailu.

**Visualization:** Hordofa Gutema, Iydia Melkamu, Natnael Kebede, Elias Assefa, Addis Hordofa.

**Writing – original draft:** Muluken Yigezu.

**Writing – review & editing:** Muluken Yigezu, Hordofa Gutema, Iydia Melkamu, Natnael Kebede, Elias Assefa, Addis Hordofa, Gebremedhin Hailu.

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
