## [Decision Letter · Decision Letter 0]

10 Dec 2024

PONE-D-24-13630Self-Care Practice And Associated Factors Among Hypertensive Patients Who Have Follow-Ups In Public Hospitals Of Bahir-Dar City, Northwest Ethiopia, A Mixed StudyPLOS ONE

Dear Dr. Yigezu,

Thank you for submitting your manuscript to PLOS ONE. After careful consideration, we feel that it has merit but does not fully meet PLOS ONE’s publication criteria as it currently stands. Therefore, we invite you to submit a revised version of the manuscript that addresses the points raised during the review process.

We look forward to receiving your revised manuscript.

Kind regards,

Blessing Onyinye Ukoha-kalu, B.Pharm, M.Pharm, Ph.D

Academic Editor

PLOS ONE

2. In the online submission form, you indicated that [Data will be available upon request from the corresponding author.]. All PLOS journals now require all data underlying the findings described in their manuscript to be freely available to other researchers, either 1. In a public repository, 2. Within the manuscript itself, or 3. Uploaded as supplementary information. This policy applies to all data except where public deposition would breach compliance with the protocol approved by your research ethics board. If your data cannot be made publicly available for ethical or legal reasons (e.g., public availability would compromise patient privacy), please explain your reasons on resubmission and your exemption request will be escalated for approval.

Additional Editor Comments (if provided):

Reviewers' comments:

Reviewer's Responses to Questions

**Comments to the Author**

1. Is the manuscript technically sound, and do the data support the conclusions?

Reviewer #1: Yes

Reviewer #2: Yes

2. Has the statistical analysis been performed appropriately and rigorously? 

Reviewer #1: Yes

Reviewer #2: Yes

3. Have the authors made all data underlying the findings in their manuscript fully available?

Reviewer #1: Yes

Reviewer #2: Yes

4. Is the manuscript presented in an intelligible fashion and written in standard English?

Reviewer #1: Yes

Reviewer #2: Yes

5. Review Comments to the Author

Reviewer #1: Title: Self-Care Practice and Associated Factors among Hypertensive Patients Who Have

Follow-Ups In Public Hospitals Of Bahir-Dar City, Northwest Ethiopia, A Mixed Study

Manuscript Number: PONE-D-24-13630

Comment for authors

General comment

Overall the research is good but has some sort of grammatical errors specially at the abstract and discussion section

Specific comments

Method section

Eligibility criteria

It is not correct please clearly specify your inclusion criteria

Sampling method

The sampling method for qualitative part is purposive but the samples were selected by considering some factors but it seems qota sampling

Please rethink over it because reader understand that it is qota sampling

Data collection

Have you checked the reliability of your data collection tool? If so how you checked?

Data analysis

The statistical significance of associations between variables was determined using odds ratios with a 95% confidence interval (CI) and p-values below 0.05

This statement is repeated please delete one of the two

Result

Instead of using tables as reference at each of statements please write in one paragraph and then use the table as reference and put it once

Discussion

The qualitative part of your finding is not discussed well please raise key issues and discuss with literatures

Declaration

Ethical consideration

Informed voluntary written and signed consent was obtained from all study participants prior to start data collection

why you choose written consent?

Reviewer #2: This manuscript is well-written, it could make a valuable contribution to literature on self-care practices of hypertension.

In result, it would be better if minimum and maximum age is also included along with average age of the participants

Thank you once again for your contributions.

6. PLOS authors have the option to publish the peer review history of their article (what does this mean? ). If published, this will include your full peer review and any attached files.

**Do you want your identity to be public for this peer review?** For information about this choice, including consent withdrawal, please see our Privacy Policy .

Reviewer #1: No

Reviewer #2: **Yes: ** Santoshi Adhikari

---

## [Author Response · Author response to Decision Letter 1]

2 Jan 2025

Author response to reviewers

Submission ID PONE-D-24-13630

MANUSCRIPT Title: Self-Care Practice And Associated Factors Among Hypertensive Patients Who Have Follow-Ups In Public Hospitals Of Bahir-Dar City, Northwest Ethiopia, A Mixed Study

Reviewer 1 comment:

1.Comment for authors

General comment

Overall the research is good but has some sort of grammatical errors specially at the abstract and discussion section

Thanks for your constructive comments: we have revised based on your comments

2.Specific comments

Method section

Eligibility criteria

It is not correct please clearly specify your inclusion criteria

Thanks for your constructive comments: we have revised based on your comments.

All hypertensive patients aged 18 and older who had follow-ups at public hospitals in Bahir Dar city were included in this study. However, patients who were critically ill or had cognitive impairments during the data collection period were excluded. Additionally, patients with follow-up periods of less than six months were excluded, as a certain activity or behavior is assumed to be established only if it is practiced repeatedly for at least six months.

3.Sampling method

The sampling method for qualitative part is purposive but the samples were selected by considering some factors but it seems qota sampling

Please rethink over it because reader understand that it is qota sampling

Thanks for your constructive comments: we have revised based on your comments.

Participants in the qualitative study was selected purposively on a quota basis.

4.Data collection

Have you checked the reliability of your data collection tool? If so how you checked?

Thank you for your insightful comment.

We have checked reliability test for our tool, to assess reliability of the items individual Cronbach's Alpha was computed for individual latent variables and overall items of the tool. The overall Cronbach's Alpha consisting of 70 items was 0.902. The Cronbach's Alpha for the latent variables is also presented in table 6.

5.Data analysis

The statistical significance of associations between variables was determined using odds ratios with a 95% confidence interval (CI) and p-values below 0.05

This statement is repeated please delete one of the two

Thanks for your constructive comments: we have revised based on your comments.

6.Result

Instead of using tables as reference at each of statements please write in one paragraph and then use the table as reference and put it once

Thanks for your constructive comments: we have revised based on your comments.

7.Discussion

The qualitative part of your finding is not discussed well please raise key issues and discuss with literatures

Thanks for your constructive comments: we have revised based on your comments.

8. Declaration

Ethical consideration

Informed voluntary written and signed consent was obtained from all study participants prior to start data collection

why you choose written consent?

Thank you for your insightful comment.

The preference for informed voluntary written and signed consent from study participants over oral consent is based on several key reasons:

1. Documentation: Written consent provides a clear record of the participant's agreement, serving as legal and ethical evidence that they were informed and willingly participated.

2. Clarity and Understanding: It allows researchers to communicate study details clearly, ensuring participants fully understand what they are agreeing to, which is essential for informed consent.

3. Legal Protection: Written consent includes language that safeguards participants' rights, including their ability to withdraw, confidentiality, and potential compensation.

4. Accountability: Requiring written consent encourages participants to engage actively and make conscious decisions about their involvement, enhancing the quality of research outcomes.

5. Ethical Considerations: Obtaining written consent aligns with ethical research principles by ensuring participants are fully informed and voluntarily consenting, thus maintaining research integrity.

While oral consent may be appropriate in emergencies, written consent is generally preferred to adequately inform participants and protect their rights.

Reviewer 2:

1.This manuscript is well-written, it could make a valuable contribution to literature on self-care practices of hypertension.

In result, it would be better if minimum and maximum age is also included along with average age of the participants

Thank you once again for your contributions.

Thanks for your constructive comments: we have revised based on your comments. The participants in the study ranged in age from a minimum of 25 years to a maximum of 89 years, with a mean age of 53.52 ± 13.7 years (standard deviation).

---

## [Editor Report · Decision Letter 1]

8 Jan 2025

Self-Care Practice And Associated Factors Among Hypertensive Patients Who Have Follow-Ups In Public Hospitals Of Bahir-Dar City, Northwest Ethiopia, A Mixed Study

PONE-D-24-13630R1

Dear Dr. Yigezu,

We’re pleased to inform you that your manuscript has been judged scientifically suitable for publication and will be formally accepted for publication once it meets all outstanding technical requirements.

Kind regards,

Blessing Onyinye Ukoha-kalu, B.Pharm, M.Pharm, Ph.D

Academic Editor

PLOS ONE

Additional Editor Comments: Thank you for taking the time to address the reviewers' concerns.
---

## [Editor Report · Acceptance letter]

PONE-D-24-13630R1

PLOS ONE

Dear Dr. Yigezu,

I'm pleased to inform you that your manuscript has been deemed suitable for publication in PLOS ONE. Congratulations! Your manuscript is now being handed over to our production team.

Kind regards,

on behalf of

Dr Blessing Onyinye Ukoha-kalu

Academic Editor

PLOS ONE